# Volatile Organic Compounds from *Bacillus aryabhattai* MCCC 1K02966 with Multiple Modes against *Meloidogyne incognita*

**DOI:** 10.3390/molecules27010103

**Published:** 2021-12-24

**Authors:** Wen Chen, Jinping Wang, Dian Huang, Wanli Cheng, Zongze Shao, Minmin Cai, Longyu Zheng, Ziniu Yu, Jibin Zhang

**Affiliations:** 1State Key Laboratory of Agricultural Microbiology, National Engineering Research Center of Microbe Pesticides, College of Life Science and Technology, Huazhong Agricultural University, Wuhan 430070, China; chwe165431@163.com (W.C.); wjp20211206@163.com (J.W.); hui8229hd@163.com (D.H.); chengwanli@mail.hzau.edu.cn (W.C.); cmm114@mail.hzau.edu.cn (M.C.); ly.zheng@mail.hzau.edu.cn (L.Z.); yz41@mail.hzau.edu.cn (Z.Y.); 2Key Laboratory of Marine Biogenetic Resources, Third Institute of Oceanography, Ministry of Natural Resources, Xiamen 361005, China; shaozz@163.com

**Keywords:** *Bacillus aryabhattai*, *Meloidogyne incognita*, volatile organic compounds, methyl thioacetate, multiple nematicidal modes

## Abstract

Plant-parasitic nematodes cause severe losses to crop production and economies all over the world. *Bacillus aryabhattai* MCCC 1K02966, a deep-sea bacterium, was obtained from the Southwest Indian Ocean and showed nematicidal and fumigant activities against *Meloidogyne incognita* in vitro. The nematicidal volatile organic compounds (VOCs) from the fermentation broth of *B. aryabhattai* MCCC 1K02966 were investigated further using solid-phase microextraction gas chromatography-mass spectrometry. Four VOCs, namely, pentane, 1-butanol, methyl thioacetate, and dimethyl disulfide, were identified in the fermentation broth. Among these VOCs, methyl thioacetate exhibited multiple nematicidal activities, including contact nematicidal, fumigant, and repellent activities against *M. incognita*. Methyl thioacetate showed a significant contact nematicidal activity with 87.90% mortality at 0.01 mg/mL by 72 h, fumigant activity in mortality 91.10% at 1 mg/mL by 48 h, and repellent activity at 0.01–10 mg/mL. In addition, methyl thioacetate exhibited 80–100% egg-hatching inhibition on the 7th day over the range of 0.5 mg/mL to 5 mg/mL. These results showed that methyl thioacetate from MCCC 1K02966 control *M. incognita* with multiple nematicidal modes and can be used as a potential biological control agent.

## 1. Introduction

Plant-parasitic nematodes (PPNs) cause a serious threat to a broad range of plants and agricultural crops all over the world [1]. So far, more than 4100 species of PPNs and more than 5500 species of host plants have been reported [2,3]. They can parasitize on almost every vascular plant [4]. It has been estimated that PPNs caused about $100–$150 billion in annual economic losses to agriculture, more than half of them were caused by *Meloidogyne* spp., which are widely distributed in multiple transmission routes with strong drug resistance and are difficult to prevent and control [5,6]. At present, chemical nematicides are still the mainstream of root-knot nematode control. With the improvement of food security and environmental protection requirements, highly toxic chemical nematicides are no longer in line with the sustainable development strategy of modern agriculture. Numerous highly toxic nematicides, such as bromomethane and ethoprophos, are beginning to be included in phase-out plans or banned [7,8,9]. Biological control has been shown economically and ecologically beneficial to reduce the losses caused by nematodes, and the microbial preparations with nematicidal activity and environmental friendliness have become the focus of nematode control. A number of studies have shown that *Bacillus*, *Pseudomonas*, and *Pasteurella* have great potential in the control of PPNs [10,11,12].

Although a number of nematicides have been successfully extracted from terrestrial microorganisms, the lack of drug classes and the current challenge of drug resistance highlight the importance of discovering new natural products with biological activity against PPNs from natural resources [13,14]. The ocean has a unique ecological environment; these unique environmental conditions enable marine microorganisms to metabolise several different organic molecules to obtain extremely limited resources, and these small molecules provide rich resources for the development of marine biological active substances [15]. Therefore, screening new strains from marine microorganisms and extracting their disease-resistant and pesticidal active substances for agricultural biological control can gradually slow down or eliminate the serious harm to the environment caused by chemical control, which is also the focus of important biological control research in the direction of plant protection [16,17,18].

*Bacillus aryabhattai* is a plant growth-promoting bacterium first reported in 2012 [19]. As a rhizosphere microorganism, *B. aryabhattai* has a variety of biological activities. Chen et al. [20] isolated a strain of *B. aryabhattai* SK1-7 from the rhizosphere soil of poplar; the bacterium showed a certain ability to dissolve potassium and a good growth promotion effect on poplar. A strain of *B. aryabhattai* AIS-10 was isolated from the rhizosphere soil of saffron and played an important role in plant growth-promoting properties, such as dissolving phosphate, producing iron carrier, hydrocyanic acid, and antagonizing plant pathogenic fungi in vitro [21]. *B. aryabhattai* MoB09 can degrade the residual chemical herbicide Paraquat in soil and promote plant growth [22]. Therefore, *B. aryabhattai*, as a plant growth-promoting rhizobacteria is an ideal nematicide and has a broad application prospect.

Microbial pesticides are mostly live bacterial preparations, their colonization and production of active substances may be affected by a variety of environmental conditions in the field, resulting in unsatisfactory control effects. Therefore, one of the measures to solve the stability of microbial pesticides is to isolate nematicidal active substances from secondary metabolites of strains [23]. Volatile organic compounds (VOCs) are a kind of lipophilic compound with low molecular weight, high vapor pressure, and low boiling point [24], and they widely exist in plants and microorganisms; a number of natural VOCs have attracted the attention of researchers at home and abroad because of their advantages of preventing plant diseases and improving crop yield [25]. Currently, VOCs with biological activities are mainly produced by fungi and bacteria. Cheng et al. [26] isolated eleven volatile organic compounds from the strain KM2501-1, of which 8 had contact nematicidal activity, 6 had fumigation activity, and 5 had stable chemotaxis to root-knot nematodes. Terra et al. [27] observed that VOCs, such as 2-methyl butyl acetate, methyl butyl acetate, ethyl acetate, and 2-methyl propyl acetate from the *Fusarium oxysporum* strain 21 had excellent nematicidal activity against *M. incognita*.

*B. aryabhattai* MCCC 1K02966 was isolated at a water depth of 2700 m in the Southwest Indian Ocean. In this study, MCCC 1K02966 exhibited nematicidal activity against *M. incognita*. The VOCs produced by MCCC 1K02966 were isolated and identified, and their nematicidal activities were evaluated. These results showed that methyl thioacetate control *M. incognita* with multiple nematicidal modes.

## 2. Results

### 2.1. Nematicidal Activities of B. aryabhattai MCCC 1K02966 Culture Filtrates against M. incognita

The culture filtrates of *B. aryabhattai* MCCC 1K02966 showed a contact nematicidal activity against *M. incognita* in vitro. The mortality rate was 71.56% when the nematode juveniles were exposed to the culture filtrates of strain MCCC 1K02966 at 72 h, whereas the mortality in the 2216E medium (CK) was 8.20% (Figure 1A). In addition, the culture filtrates of MCCC 1K02966 also exhibited fumigant activity against *M. incognita*, with a mortality rate of 55.38% at 72 h (Figure 1B)., indicating that the strain MCCC 1K02966 can produce nematicidal volatiles.

### 2.2. Identification of VOCs of B. aryabhattai MCCC 1K02966

The VOCs produced by the fermentation broth of strain MCCC 1K02966 were analysed and identified by gas chromatography-mass spectrometry (GC-MS). Seven peaks from the fermentation broth of MCCC 1K02966 were observed in the total ion current chromatograms (Figure 2). After removing the background substances in the 2216E medium, four VOCs from the fermentation broth of MCCC 1K02966, namely, pentane, 1-butanol, disulfide dimethyl, and methyl thioacetate, were screened (Table 1).

### 2.3. Nematicidal Activity of VOCs against M. incognita

In order to verify the speculation that the nematicidal VOCs are the primary nematicidal toxin of strain MCCC 1K02966, four VOCs produced by *B. aryabhattai* MCCC 1K02966 were purchased (Table 1). Among these VOCs, dimethyl disulfide has been reported to have nematicidal activity [28,29,30], this research focused on the nematicidal activity of pentane, 1-butanol, and methyl thioacetate. Second-stage juveniles (J2s) were immersed in these VOCs at various concentrations to detect contact nematicidal activity against *M. incognita*. Methyl thioacetate showed 85–100% mortality of the contact nematicidal activity over a range from 0.01 mg/mL to 1 mg/mL at 72 h (Table 2). The mortality rates of 1-butanol and pentane against *M. incognita* were below 10% at a concentration of 1 mg/mL.

### 2.4. Fumigant Activity of Methyl Thioacetate against M. incognita

The fumigant activity of methyl thioacetate against *M. incognita* was detected in 96-well plates. The results indicated that methyl thioacetate had a significant fumigant activity against *M. incognita*. Methyl thioacetate showed above 95% mortality of the fumigant activity over a range from 1 mg/mL to 10 mg/mL at 72 h (Table 3).

### 2.5. Egg-Hatching Inhibition Activity of Methyl Thioacetate against M. incognita

A number of egg masses were used to detect the egg-hatching of *M. incognita* immersed in methyl thioacetate at various concentrations. The results showed that the hatching number of nematodes was inhibited in all treatment groups and more than 80–100% egg-hatching inhibition on the seventh day over a dose range of 0.5 mg/mL to 5 mg/mL (Figure 3). Meanwhile, when methyl thioacetate was tested for the inhibition of egg-hatching, distinct dose-response relationships and significant inhibition of egg-hatching were evident after seven days of exposure.

### 2.6. Chemotaxis of M. incognita towards Methyl Thioacetate

Using a population chemotaxis assay, we tested the J2s of *M. incognita* for responses to methyl thioacetate at concentrations over the range of 0.01 mg/mL to 10 mg/mL. If the chemotaxis indexes (C.I.) of four concentrations were significantly different compared with the C.I. of the control (0 mg/mL), then the activity of methyl thioacetate influenced the chemotaxis of J2s. The results (Figure 4) showed that methyl thioacetate treatment had a C.I. of −0.22, −0.19, −0.16, and −0.09 from high to low concentrations, and the C.I. of methyl thioacetate significantly differed compared with those of the control at 0.1–10 mg/mL. Methyl thioacetate had a repellent activity against *M. incognita*.

## 3. Discussion

In recent years, people have gradually paid attention to environmental protection, and a series of highly efficient, broad-spectrum, and low-toxicity nematicides has arrived at the market. However, nematode resistance gradually increased due to the frequent or excessive use of drugs. Thus, the application of a single nematicide or a single nematicidal function promotes the development of resistance, which makes nematode control more difficult [31,32]. Therefore, screening of environmentally friendly strains and active substances is an effective way to solve the drug resistance of nematodes. *B. aryabhattai* MCCC 1K02966 was isolated at a water depth of 2700 m in the Southwestern Indian Ocean. This strain has a dual nematicidal activity against root-knot nematodes by contact and fumigation (Figure 1) and provides a valuable marine microbial resource for nematode control.

Microorganisms can release volatile substances in soil or other growth substrates to prevent and control plant pathogenic nematodes through multiple modes [33]; the volatile substances are considered potential molecules for the development of commercial nematicides [25]. *B. aryabhattai* MCCC 1K02966 has nematicidal activity in vitro (Figure 1). In this study, the VOCs produced by the fermentation broth of MCCC 1K02966 were isolated and identified; four VOCs, namely, pentane, dimethyl disulfide, 1-butanol, and methyl thioacetate, were analysed by GC-MS (Figure 2 and Table 1). Among these VOCs, dimethyl disulfide has been reported as an alternative to traditional fumigants for root-knot nematodes control in 2010 [34]. Dimethyl disulfide has contact and fumigant activity against root-knot nematodes [28] and an attractive activity on them [35]. Moreover, dimethyl disulfide as a fumigant was effective against root-knot nematodes in greenhouse and field experiments [29,30,34]. Therefore, we purchased the commercial products of the other three VOCs for bioassay. We observed that 1-butanol and pentane had no nematicidal activity, but 1-butanol showed an attractive activity on root-knot nematodes at low concentrations [36]. We mainly studied the nematicidal activity of methyl thioacetate and observed that methyl thioacetate showed a significant contact nematicidal activity with 87.90% mortality at 0.01 mg/mL by 72 h (Table 2), fumigant activity in mortality 91.10% at 1 mg/mL by 48 h (Table 3) and 80–100% inhibition on the nematode eggs on the 7th day over the range of 0.5 mg/mL to 5 mg/mL (Figure 3). Meanwhile, the contact nematicidal and fumigant activities of methyl thioacetate against *M. incognita* at 72 h were equivalent to that of furfural acetone (Table 2 and Table 3), which has been reported to have significant nematicidal activities [26]. In addition, methyl thioacetate exhibited a repellent activity at 0.01–10 mg/mL (Figure 4). These results showed that methyl thioacetate from the fermentation broth of *B. aryabhattai* MCCC 1K02966 has multiple nematicidal activities, including contact nematicidal activity, fumigant activity, inhibition of egg hatching and repellent activity. This study is the first report that methyl thioacetate has significant multiple nematicidal activities.

Methyl thioacetate is an important sulfur-containing spice with strong onion, garlic, and radish-like aroma and is commonly used in various flavourings. Methyl thioacetate is an important aroma component of cheese; eight strains of *Brevibacterium flax* were isolated from cheese. Among these strains, four strains can produce methyl thioacetate, which is the first demonstration that microorganisms can produce such a compound [37]. Methyl thioacetate is also a flavour substance in beer and contributes significantly to the flavour profile of certain beers [38]. Hence, methyl thioacetate is a low toxic and environmentally friendly active substance. Methyl thioacetate can cooperate with dimethyl trisulfide to induce carrion beetles for humus decomposition [39]. Paolina et al. observed that methyl thioacetate had antifungal activity, but not antibacterial activity and stimulated the growth of several bacteria [40]. Thus far, the role of methyl thioacetate in the prevention and control of nematodes has not been reported. In this study, we observed for the first time that methyl thioacetate has multiple modes against root-knot nematodes. Compared with traditional nematicides with a single activity, the greatest advantage of methyl thioacetate is the highly effective multiple nematicidal activities. With its good dispersibility and penetration into the soil as a VOC, it can inhibit or kill plant pathogenic nematodes and eggs in the soil around plant roots through multiple pathways to achieve plant protection. As a nematicide, methyl thioacetate may also inhibit the growth of phytopathogenic fungi and promote the growth of plant growth-promoting rhizobacteria when applied in the field. Therefore, as a potential environmentally friendly nematicide, methyl thioacetate has a broad application potential in current agriculture. We can combine the multiple nematicidal activities of methyl thioacetate to establish a comprehensive strategy to control nematodes, and the specific implementation scheme needs to be further studied in the field.

## 4. Materials and Methods

### 4.1. Bacterial Materials and Chemicals

*B. aryabhattai* MCCC 1K02966 was provided from the Third Institute of Oceanography, Ministry of Natural Resources (Xiamen, China), which was isolated from Southwestern Indian Ocean (water depth of 2700 m). The bacterium was cultured in artificial seawater medium 2216E at 28 °C and 180 r/min for 48 h (OD600 = 1.8) to detect nematicidal activity and analyse nematicidal substances.

The following chemicals were used: methyl thioacetate (>95%; Macklin, Shanghai, China), pentane (>99%; Macklin, Shanghai, China), 1-butanol (>99%; Macklin, Shanghai, China), and ethanol (>99.7, Biosharp, Shanghai, China).

### 4.2. Collection of M. incognita Egg Masses and J2s

*A population of* egg masses was collected from the roots with severe *Meloidogyne incognita* infected tomato plants (*Solanum lycopersicum* L.). The roots were gently rinsed to remove the attached soil, and excess water from the roots was wiped away. Egg masses were picked for the inhibition of egg hatching experiments or collected into 24-well plates with sterile water for incubation. J2s can be collected after three to five days to prepare a suspension for nematicidal experiments.

### 4.3. Nematicidal Activity In Vitro of Culture Filtrates of B. aryabhattai MCCC 1K02966 against M. incognita

The culture filtrates of strain MCCC 1K02966 were prepared. To examine nematicidal activity, 120 µL undiluted culture filtrate was added to a 96-well plate and each well was filled with a freshly incubated suspension of approximately 40 J2s. The 2216E medium was used as the CK. Each treatment was replicated thrice. The plates were immediately wrapped with parafilm and maintained in the dark at 20 °C. The number of dead nematodes was observed at 48 and 72 h under an inverted microscope. *M. incognita* was considered dead when no movement was observed for 2 s after contact with a needle. The mortality values were corrected by eliminating natural death in a negative control using Schneider–Orelli’s formula [41].

### 4.4. Nematicidal Activity of the VOCs of B. aryabhattai MCCC 1K02966

The nematicidal activity of the VOCs produced by *B. aryabhattai* MCCC 1K02966 was detected by adding 200 µL culture filtrates to one well in the centre of a 96-well plate, and the four surrounding adjacent wells each received about 50 J2s suspended in 100 µL distilled water. The plates were immediately wrapped with parafilm to prevent the escape of volatiles. The nematodes were incubated at 20 °C for three days. The death number of nematodes was observed under an inverted microscope, and the dead nematodes were judged by a needle.

### 4.5. Identification of VOCs from B. aryabhattai MCCC 1K02966

VOCs from the fermentation broth of strain MCCC 1K02966 were extracted and identified by SPME/GC-MS [42]. The strain MCCC 1K02966 was cultured, and its VOCs were collected and identified in accordance with the methods described by Cheng et al. [26]. A new SPME fiber (75 μm *Carboxen*/*Polydimethylsiloxane*) was equilibrated with helium at 270 °C for 30 min. Extractions were performed in 20 mL Supelco SPME vials filled with 3 mL bacterial culture. The SPME needle was used to pierce the septum and the fibers are exposed to the headspace of the vial at 60 °C for 1 h with constant magnetic stirring. The VOCs from 3 mL 2216E broth were used as controls. Each treatment was repeated thrice. 

After extraction, the SPME fiber was directly inserted into the front inlet of the GC-MS instrument (Hewlett-Packard (HP) 7890A-5975C, Agilent Technologies, Santa Clara, CA, USA) and desorbed at 270 °C for 2 min. The GC/MS instrument equipped with a HP-5MS capillary column was used to separate and identify the VOCs. The carrier gas was helium with a constant flow rate of 1 mL/min. The oven temperature was programmed as follows: 40 °C for 2 min, 40 °C–180 °C at a rate of 4 °C/min, 180 °C–250 °C at 5 °C/min, and held at 240 °C for 6 min. The temperature of the transfer line and ion trap were 150 and 250 °C, respectively. The volatile organic compounds were identified from the database search through the comparison of the mass spectrum with standards in the GC/MS system data bank of the National Institute of Standards and Technology (NIST 08).4.6. Contact nematicidal activity of VOCs against M. incognita

The four main VOCs identified by GC-MS, namely, pentane, 1-butanol, methyl thioacetate, and dimethyl disulfide, were individually tested for their nematicidal activity against *M. incognita*. Test compounds solutions were prepared in ethanol and were successively diluted in distilled water. Final concentrations of ethanol in treatment wells did not exceed 2%. To examine the contact nematicidal activity in vitro, we added 120 µL commercial VOCs at various concentrations to 96-well plates and filled the wells with J2s (about 40 *M. incognita*/well). Solvent carriers were used as negative controls. Furfural acetone, which has been reported to have contact nematicidal and fumigant activities [26], was used as a positive control. The plates were immediately wrapped with parafilm and maintained in the dark at 20 °C. The number of dead nematodes was observed at 48 h and 72 h under an inverted microscope. *M. incognita* was considered dead when no movement was observed for 2 s after contact with a needle. Each treatment was replicated thrice.

### 4.6. Fumigant Activity of Methyl Thioacetate against M. incognita

A central well in each 96-well plate was added with 200 µL test methyl thioacetate solution, and the four surrounding adjacent wells were filled with approximately 50 J2s suspended in 100 µL distilled water. The plates were immediately wrapped with parafilm to prevent the escape of volatiles. The nematodes were incubated at 20 °C for 3 days. The death number of nematodes was observed under an inverted microscope, and the dead nematodes were assessed by a needle.

### 4.7. Inhibition Activity of Methyl Thioacetate on Egg Hatching of M. incognita

The method used in this experiment was similar to the contact nematicidal activity to *M. incognita*. A number of egg masses were collected from infected roots. The nematodes in each well were replaced with a single egg mass, and the solvent carrier was used as the negative control. The number of hatched nematodes was counted under the microscope after 7 days. According to the number of nematodes hatched in the negative control, the inhibition egg-hatching rate of the experimental group was corrected. Each treatment was repeated thrice.

### 4.8. Chemotaxis of M. incognita toward Methyl Thioacetate

Chemotaxis was assessed in 9 cm Petri dishes in accordance with the method described by Cheng et al. [16]. We tested the J2s of *M. incognita* for responses to methyl thioacetate at concentrations over a range from 0.01 mg/mL to 10 mg/mL. Each 9 cm Petri dish consisted of an experimental area and a control area, and the same size sterile filter papers were used to hold the test solution in the same position on both sides. The sterile filter papers in the experimental area were added with 30 µL test methyl thioacetate solution, and the 30 µL solvent carrier was added to sterile filter paper in the control area. Chemotaxis assays were performed at 20 °C for 8 h in the dark. The C.I. was calculated based on the number of nematodes in the experimental and control areas. The solvent carriers were added to sterile filter paper on both areas as the negative control.

### 4.9. Statistical Analysis

Data were analysed using SPSS, version 23.0 software (SPSS, Chicago, IL, USA) and shown as the mean ± SE (n ≥ 3). *T*-test was employed to test for significant between treatments and control, and significant differences were determined based on a threshold of * *p* < 0.05; ** *p* < 0.01 and *** *p* < 0.001. Fisher’s least significant difference (LSD) test was employed to test for significant differences between treatments, different lowercase letters indicate significant difference between treatments (*p* < 0.05).

## Figures and Tables

**Figure 1 molecules-27-00103-f001:**
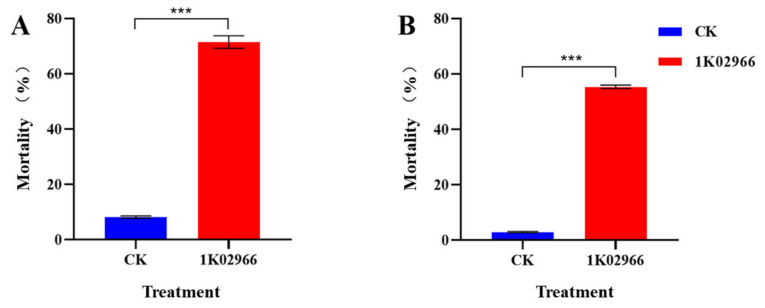
Nematicidal activity of *B. aryabhattai* MCCC 1K02966 against *M. incognita*. (**A**) Contact nematicidal activity of *B. aryabhattai* MCCC 1K02966 culture filtrate against *M. incognita* immersed in treatment wells. (**B**) Fumigant activity of *B. aryabhattai* MCCC 1K02966. The data are shown as the mean ± standard error (SE) (n ≥ 3). Statistical comparisons between the values of treatments and control were performed using a *t*-test. Significant differences were determined according to a threshold of *** *p* < 0.001.

**Figure 2 molecules-27-00103-f002:**
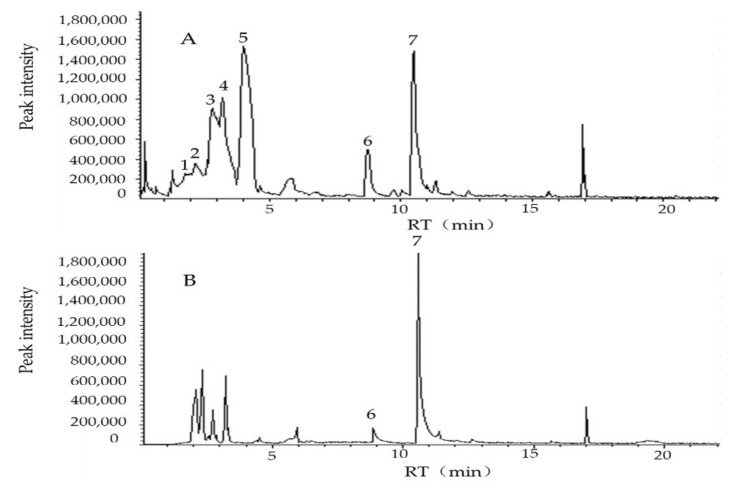
GC-MS chromatograms of (**A**) fermentation broth of MCCC 1K02966 and (**B**) 2216E. Peaks: (1) methyl thioacetate, (2) pentane, (3) 1-butanol, (4) methyl thioacetate, (5) disulfide dimethyl, (6) 2,5-dimethyl pyrazine, and (7) benzaldehyde.

**Figure 3 molecules-27-00103-f003:**
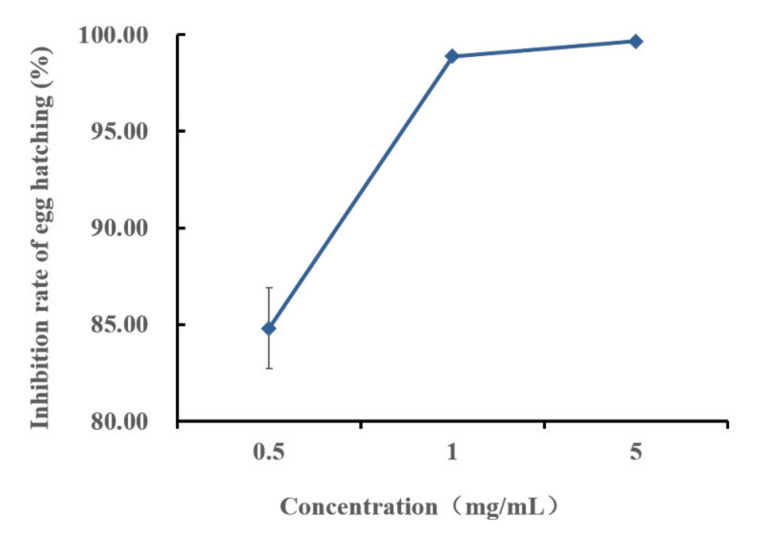
Curve of efficiency of methyl thioacetate on the egg-hatching of *M. incognita* from egg masses. Data are shown as the mean ± SE (n = 3).

**Figure 4 molecules-27-00103-f004:**
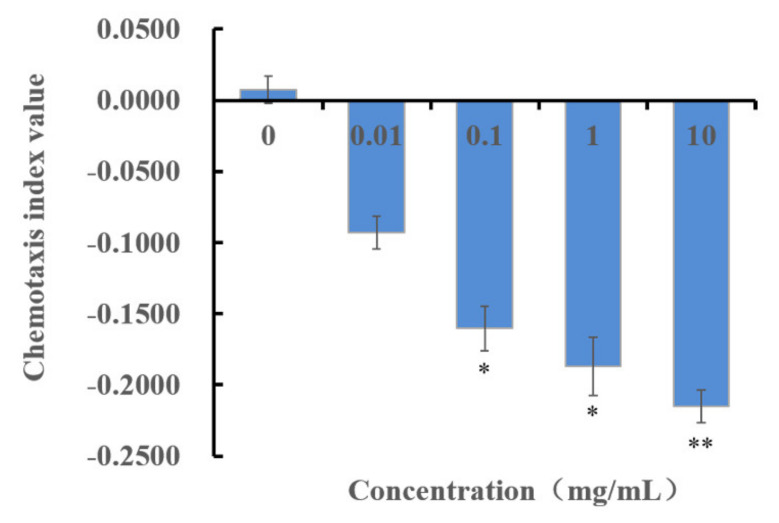
Chemotaxis of *M. incognita* towards methyl thioacetate. Data are shown as the mean ± SE (n = 6). Statistical comparisons between the values of treatments and control (0 mg/mL) were performed using a *t*-test. Significant differences were determined according to a threshold of * *p* < 0.05, ** *p* < 0.01.

**Table 1 molecules-27-00103-t001:** Area percentage of the VOCs of *B. aryabhattai* MCCC 1K02966 through GC-MS chromatogram identification.

PK	RT	RI	Area Pct	Library/ID	Formula	Molecular Weight (KD)	CAS
1	1.7781	706	2.134	Methyl thioacetate	C_3_H_6_OS	90.14	001534-08-3
2	2.2117	518	5.022	Pentane	C_5_H_12_	72.15	000109-66-0
3	2.8316	662	11.691	1-Butanol	C_4_H_10_O	74.12	000071-36-3
4	3.2174	706	12.203	Methyl thioacetate	C_3_H_6_OS	90.14	001534-08-3
5	4.0412	722	23.484	Disulfide dimethyl	C_2_H_6_S_2_	94.20	000624-92-0
6	8.7493	894	4.018	2,5-Dimethyl Pyrazine	C_6_H_8_N_2_	108.14	000123-32-0
7	10.4921	982	11.732	Benzaldehyde	C_7_H_6_O	106.12	000100-52-7

**Table 2 molecules-27-00103-t002:** Contact nematicidal activity of the VOCs against *M. incognita*.

Compounds	Concentration (mg/mL)	Mortality (%) ± SE	
		48 h	72 h
Methyl thioacetate	1	91.78 ± 1.06 c	97.28 ± 0.61 c
	0.1	83.19 ± 0.85 bc	90.92 ± 0.49 b
	0.01	78.82 ± 0.54 b	87.90 ± 0.83 b
1-Butanol	1	3.33 ± 0.71 a	5.81 ± 0.26 a
Pentane	1	2.41 ± 0.96 a	3.05 ± 0.80 a
Furfural acetone	1	100 ± 0 d	100 ± 0 c
control		0.40 ± 0.10 a	2.30 ± 0.34 a

The data are shown as the mean ± SE (n = 3). Different lowercase letters indicate significant difference between treatments (LSD test, *p* < 0.05).

**Table 3 molecules-27-00103-t003:** Fumigant activity of methyl thioacetate against *M. incognita*.

Compounds	Concentration (mg/mL)	Mortality (%) ± SE	
		48 h	72 h
Methyl thioacetate	10	97.46 ± 0.73 bc	100 ± 0 b
	5	91.87 ± 0.93 b	98.58 ± 0.82 b
	1	91.10 ± 1.09 b	97.18 ± 1.29 b
Furfural acetone	1	100 ± 0 c	100 ± 0 b
control		1.28 ± 0.46 a	1.48 ± 0.55 a

The data are shown as the mean ± SE (n = 3). Different lowercase letters indicate significant difference between treatments (LSD test, *p* < 0.05).

## Data Availability

All data contained within this article.

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
