# Peer review of "Volatile Organic Compounds from Bacillus aryabhattai MCCC 1K02966 with Multiple Modes against Meloidogyne incognita"

_molecules, 2021, doi:10.3390/molecules27010103_

Round 1

Reviewer 1 Report

The authors present a manuscript entitled  Volatile Organic Compounds from Bacillus aryabhattai MCCC 1K02966 with Multiple modes against Meloidogyne incognita. According to GCMS analysis, four VOCs were identified as pentane, 1-butanol, disulfide dimethyl, and methyl thioacetate. Direct comparison especially retention time and peak area have been done for the four VOCs. 

More complete information is better to provide in supporting information as this is the result of compound identifications.

Moreover,  this is related to their biological activity.

Another suggestion is related to control positive.

It is better to provide a control positive to compare the effectiveness of identified molecules in all performed biological assay systems.

The manuscript can be accepted after the revision.

Author Response

Dear reviewer:

Thank you for your comments on our manuscript entitled “Volatile Organic Compounds from Bacillus aryabhattai MCCC 1K02966 with Multiple Modes against Meloidogyne incognita” (Manuscript ID: molecules-1521403). These comments are very helpful for revising and improving our paper, as well as the important guiding significance to our research. We have studied your valuable comments carefully, and tried our best to revise the manuscript. The point-to-point responses to reviewers’ comments are listed as follows (the replies are highlighted in red).

The authors present a manuscript entitled  Volatile Organic Compounds from Bacillus aryabhattai MCCC 1K02966 with Multiple modes against Meloidogyne incognita. According to GC-MS analysis, four VOCs were identified as pentane, 1-butanol, disulfide dimethyl, and methyl thioacetate. Direct comparison especially retention time and peak area have been done for the four VOCs. 

1. More complete information is better to provide in supporting information as this is the result of compound identifications. Moreover,  this is related to their biological activity.

Answer: Thanks for your good suggestions. We have provided the molecular formula and molecular weight of VOCs according to your suggestion. We have added it in the revised manuscript (page 4, Table 1).

2. Another suggestion is related to control positive. It is better to provide a control positive to compare the effectiveness of identified molecules in all performed biological assay systems.

Answer: Many thanks for your kind attention and the suggestions for improving the quality of my manuscript. In the nematicidal activity assay, we have used furfural acetone, which has been reported to have nematicidal activities by our team (Cheng et al., 2017), as a positive control for comparison. We have added it in the revised manuscript (page 4, Table 2 and 3) and some content in discussion (page 6 lines 197 to 200 in red color).

Reference: Cheng, W.L.; Yang, J.Y.; Nie, Q.Y.; Huang, D.; Yu, C.; Zheng, L.Y.; Cai, M.M.; Thomashow, L.S.; Weller, D.M.; Yu, Z.N.; et al. Volatile organic compounds from Paenibacillus polymyxa KM2501-1 control Meloidogyne incognita by multiple strategies. Sci. Rep. 2017, 7, DOI: 10.1038/s41598-017-16631-8.

Reviewer 2 Report

The authors need to answer on the following questions:

1- The authors reported the use of Bacillus aryabhattai MCCC 1K02966, a deep-sea bacterium collected from Southwest Indian Ocean. The authors did not cite the article that described this bacteria? Does  this type of bacteria did not obtained previously? A background is required about this bacterial species.

2- The authors did not mention about the kind of crops that was infected by nemtode. What is the targeted crop in this paper and from which crop they collected the eggs?

3- Some detected plagiarism was observed. Some phrases showed similarity with the following citation; 

Zhai Y, Shao Z, Cai M, Zheng L, Li G, Huang D, Cheng W, Thomashow LS, Weller DM, Yu Z and Zhang J (2018) Multiple Modes of Nematode Control by Volatiles of Pseudomonas putida 1A00316 from Antarctic Soil against Meloidogyne incognitaFront. Microbiol. 9:253. doi: 10.3389/fmicb.2018.00253.

The authors need to check the similarity index for the submitted article.

4- The authors stated that "Microbial pesticides are mostly live bacterial preparations, and the control effect in the field is often affected by the strain themselves and the external environment." This sentence needed to re-written because there is an grammar error in this sentence. In addition, the introduction section should be re-written to avoid similarity in using the sentences that was stated in another article as mentioned in "query 3"

5- In results section. What is the used control in this study? Negative control and positive control should be used to estimate the efficiency of methyl thioacetate in comparison with known nematicide.  

6- Why does the authors did not showed the activity of dimethyl disulfide in this study, especially this secondary metabolite is one of the effective compound of nematode effect when they used the VOCs compounds.

7- How does the authors extracted the VOCs compounds from MCCC 1K02966 to be analyzed in GC-MS? Did they use water extract or alcohol extract? 

Author Response

Dear reviewer:

Thank you for your comments on our manuscript entitled “Volatile Organic Compounds from Bacillus aryabhattai MCCC 1K02966 with Multiple Modes against Meloidogyne incognita” (Manuscript ID: molecules-1521403). These comments are very helpful for revising and improving our paper, as well as the important guiding significance to our research. We have studied your valuable comments carefully, and tried our best to revise the manuscript. The point-to-point responses to reviewers’ comments are listed as follows (the replies are highlighted in red).

The authors need to answer on the following questions:

1. The authors reported the use of Bacillus aryabhattai MCCC 1K02966, a deep-sea bacterium collected from Southwest Indian Ocean. The authors did not cite the article that described this bacteria? Does this type of bacteria did not obtained previously? A background is required about this bacterial species.

Answer: We sincerely thank the reviewer for careful reading. Bacillus aryabhattai MCCC 1K02966 was provided by the Third Institute of Oceanography, Ministry of Natural Resources (Xiamen, China). Our team has a project collaboration with the Third Institute of Oceanography, Ministry of Natural Resources ( Project funding: China Ocean Mineral Resources Research and Development Association (DY135-B2-17)). So far, this is the first time to report this strian, and the background information about this bacterial species we have presented in the introduction of the revised manuscript (page 2 lines 61 to 71 in red color).

2. The authors did not mention about the kind of crops that was infected by nemtode. What is the targeted crop in this paper and from which crop they collected the eggs?

Answer: Many thanks for your kind attention and the suggestions for improving the quality of my manuscript. We collected the eggs masses from the roots with severe Meloidogyne incognita infected tomato plants (Solanum lycopersicum L.). We have added it in the revised manuscript (page 7, lines 241 to 242 in red color).

3. Some detected plagiarism was observed. Some phrases showed similarity with the following citation; 

Zhai Y, Shao Z, Cai M, Zheng L, Li G, Huang D, Cheng W, Thomashow LS, Weller DM, Yu Z and Zhang J (2018) Multiple Modes of Nematode Control by Volatiles of Pseudomonas putida 1A00316 from Antarctic Soil against Meloidogyne incognita. Front. Microbiol. 9:253. doi: 10.3389/fmicb.2018.00253.

The authors need to check the similarity index for the submitted article.

Answer: We sincerely thank the reviewer for careful reading and reminder. This article is also published by our team, which belongs to the same corresponding author. The language style may be more similar in writing. We have revised it in the introduction (page 1, The words in red color).

4. The authors stated that "Microbial pesticides are mostly live bacterial preparations, and the control effect in the field is often affected by the strain themselves and the external environment." This sentence needed to re-written because there is an grammar error in this sentence. In addition, the introduction section should be re-written to avoid similarity in using the sentences that was stated in another article as mentioned in "query 3".

Answer: Many thanks for your kind attention and the suggestions for improving the quality of my manuscript. We have re-written this sentence (page 2 lines 72 to 74 in red color) and the introduction section in the revised manuscript.

5. In results section. What is the used control in this study? Negative control and positive control should be used to estimate the efficiency of methyl thioacetate in comparison with known nematicide. 

Answer: Thanks for your good suggestions. In the nematicidal activity assay, we have used ethanol to prepare the test solutions, and the final concentrations of ethanol never exceeded 2%. So we have used 2% ethanol solvent carriers as the negative control. The furfural acetone, which has been reported to have nematicidal activities (Cheng et al., 2017), as a positive control for comparison. We have added it in the revised manuscript (page 4, Table 2 and 3) and some content in discussion (page 6 lines 197 to 200 in red color).

Reference: Cheng, W.L.; Yang, J.Y.; Nie, Q.Y.; Huang, D.; Yu, C.; Zheng, L.Y.; Cai, M.M.; Thomashow, L.S.; Weller, D.M.; Yu, Z.N.; et al. Volatile organic compounds from Paenibacillus polymyxa KM2501-1 control Meloidogyne incognita by multiple strategies. Sci. Rep. 2017, 7, DOI: 10.1038/s41598-017-16631-8.

6. Why does the authors did not showed the activity of dimethyl disulfide in this study, especially this secondary metabolite is one of the effective compound of nematode effect when they used the VOCs compounds.

Answer: We sincerely thank the reviewer for careful reading. Dimethyl disulfide has been reported to have nematicidal activities by our team (Huang et al., 2020), and its nematicidal activities have been described in the discussion section of the manuscript (page 6 lines 185 to 190 in red color). Therefore, the results section of the manuscript does not specifically show the nematicidal activities of dimethyl disulfide.

Reference: Huang, D.; Yu, C.; Shao, Z.Z.; Cai, M.M.; Li, G.Y.; Zheng, L.Y.; Yu, Z.N.; Zhang, J.B. Identification and characterization of nematicidal volatile organic compounds from deep-sea virgibacillus dokdonensis MCCC 1A00493. Molecules 2020, 25, DOI: 10.3390/molecules25030744.

7. How does the authors extracted the VOCs compounds from MCCC 1K02966 to be analyzed in GC-MS? Did they use water extract or alcohol extract? 

Answer: We sincerely thank the reviewer for careful reading. We did not extract the VOCs using water or ethanol. The VOCs from the fermentation broth of strain MCCC 1K02966 were extracted by solid-phase microextraction (SPME). A new SPME fiber (75 μm Carboxen/Polydimethylsiloxane) was equilibrated with helium at 270 °C for 30 min. Extractions were performed in 20 mL Supelco SPME vials filled with 3 mL bacterial culture. The SPME needle was used to pierce the septum and the fibers are exposed to the headspace of the vial at 60 °C for 1 h with constant magnetic stirring. The specific method of VOCs extraction and identification has been added to the revised manuscript (page 8 lines 269 to 284 in red color).

Round 2

Reviewer 1 Report

Dear Authors,

After revision, I recommend your manuscript accepts in the presence form.

Author Response

Dear reviewer:

Thanks for your suggestions for improving the quality of my manuscript. On behalf of my co-authors, we would like to express our great appreciation to you.

Thank you and best regard.

Reviewer 2 Report

The paper can be published in Molecules

Author Response

(The authors gave the same response as above.)
